# In-Vivo Quantification of Knee Deep-Flexion in Physiological Loading Condition trough Dynamic MRI

**Michele Conconi [1,\*]**, **Filippo De Carli [2]**, **Matteo Berni [3]**, **Nicola Sancisi [1]**, **Vincenzo Parenti-Castelli [1]** and **Giuseppe Monetti [2]**

[1] Department of Industrial Engineering—DIN, University of Bologna, 40139 Bologna, Italy
[2] Primus Forlì Medical Center, 47121 Forli, Italy
[3] Medical Technology Laboratory, IRCCS Istituto Ortopedico Rizzoli, 40139 Bologna, Italy
[\*] Correspondence: michele.conconi@unibo.it

**Abstract:** The in-vivo quantification of knee motion in physiological loading conditions is paramount for the understanding of the joint's natural behavior and the comprehension of articular disorders. Dynamic MRI (DMRI) represents an emerging technology that makes it possible to investigate the functional interaction among all the joint tissues without risks for the patient. However, traditional MRI scanners normally offer a reduced space of motion, and complex apparatus are needed to load the articulation, due to the horizontal orientation of the scanning bed. In this study, we present an experimental and computational procedure that combines an open, weight-bearing MRI scanner with an original registration algorithm to reconstruct the three-dimensional kinematics of the knee from DMRI, thus allowing the investigation of knee deep-flexion under physiological loads in space. To improve the accuracy of the procedure, an MR-compatible rig has been developed to guide the knee flexion of the patient. We tested the procedure on three volunteers. The overall rotational and positional accuracy achieved are $1.8° \pm 1.4$ and $1.2$ mm $\pm 0.8$, respectively, and they are sufficient for the characterization of the joint behavior under load.

**Keywords:** Dynamic MRI; weight-bearing MRI; knee deep-flexion

## 1. Introduction

Musculoskeletal disorders are the second most common cause of disability worldwide, exceeded only by traffic-related injuries, and they are responsible for the 21.3% of the total years lived with disability [1,2]. Among human articulations, the knee is one of the most susceptible to ligament injuries and to the risk of osteoarthritis development [3]. Understanding and identifying a patient's normal and pathological joint function is, therefore, a high clinical priority.

Static morphological imaging helps the diagnosis and the etiology identification of these disorders [4]. However, the functional characterization of musculoskeletal system in physiological conditions still relies on clinician experience [5]. Indeed, the relation between anatomical structures that can be observed during static imaging may significantly differ from what is measured during dynamic musculoskeletal tasks [6–8]. Several studies showed that evaluating a patient by means of static, non-weight-bearing scans alone may result in misdiagnoses [6–10]. In-vivo imaging of joint motion may fill the gap, providing a tool to better understand the normal joint physiology, investigating the etiology of musculoskeletal diseases, and designing more effective treatments.

Currently, in-vivo analysis of articular motion can be performed by several techniques [11]: ultrasonography [12], fluoroscopy [13], computed tomography (CT) [14], and Magnetic Resonance Imaging (MRI) [6]. Ultrasonography, however, is limited to the evaluation of soft tissues around the joint. Fluoroscopy and CT expose the patient to ionizing radiation and do not allow the direct observation of soft tissues. On the other hand, MRI

returns information of both bones and soft tissues without known risk for the patient. This, together with the recent advances in dynamic sequences, boosted the application of Dynamic MRI (DMRI) to the investigation of joint behavior [15].

The goal of this work is to present an experimental-computational procedure for the investigation of the knee deep-flexion under physiological loads. The procedure reconstructs the spatial kinematics of the knee from dynamic planar MR images. To this aim, we employed a weight-bearing MR scanner, in combination with a custom MR compatible rig, to guide the knee flexion during the dynamic scan. Finally, we developed a new registration algorithm to reconstruct the three-dimensional tibio-femoral kinematics from DMRI.

## 2. Materials and Methods

### 2.1. Experimental Setup

MR scans were performed with a 0.25 T G-Scan, Esaote SpA. Despite the low magnetic field, this scanner has the advantages of a rotatable bore (Figure 1), allowing for the weight-bearing imaging of the patient. Additionally, the scanner is open, thus making a wider mobility of the patient possible. Loaded knee flexion was performed with the scanner in vertical position and with the aid of an MR-compatible rig specifically designed to guide the knee flexion. The rig is in plastic and consists of a hydraulic step that can be lowered at controlled velocity by regulating the liquid flow from the piston sustaining the step to the accumulation tank (Figure 2). Connectors in the hydraulic circuit were made out of brass to minimize the magnetic field distortion, while the steel-controlling valves were positioned outside the scanner room and controlled by an operator.

During the scans, the volunteers stood with the right leg in the scanner, while the contralateral leg was supported by the step (Figure 2). Lowering the step, the right leg flexed under the weight of the volunteer, resulting in a physiological load comparable with what is experienced during stair climbing.

### 2.2. Preliminary Static Acquisitions

We analyzed three volunteers (age: $29 \pm 7.9$ years; height: $174.3 \pm 7.6$ cm; weight: $71.7 \pm 7.6$ kg). For each volunteer, an initial MRI of the knee (3D hybrid contrast enhancement, FOV $512 \times 512$, pixel spacing 0.5/0.5, slice thickness 0.5 mm, TR = 10 ms, TE = 5 ms, flip angle $60°$, hereinafter 3D HYCE) was taken in a supine, non-weight-bearing configuration to provide a reference image for the segmentation of all the main knee structures. In particular, bone models of the femur, tibia, and fibula were segmented through the open software MITK. Anatomical reference systems for the femur and tibia were defined based on the convention proposed by Tashman and co-workers [16], for which x, y, and z are axes respectively pointing anteriorly, proximally and to the right. For the aims of the present study, fibula and tibia were considered as a rigid complex.

### 2.3. Registration Algorithm

With the employed scanner, DMRI results in a series of subsequent planar acquisitions taken while the subject is moving. More than one plane can be scanned for the same joint pose: for instance, as clarified below, we chose to use two synchronized planes in this case, although the number could be higher. Joint kinematics can, thus, be reconstructed by registering a 3D model of the moving objects on the DMRI planes for each measured joint pose.

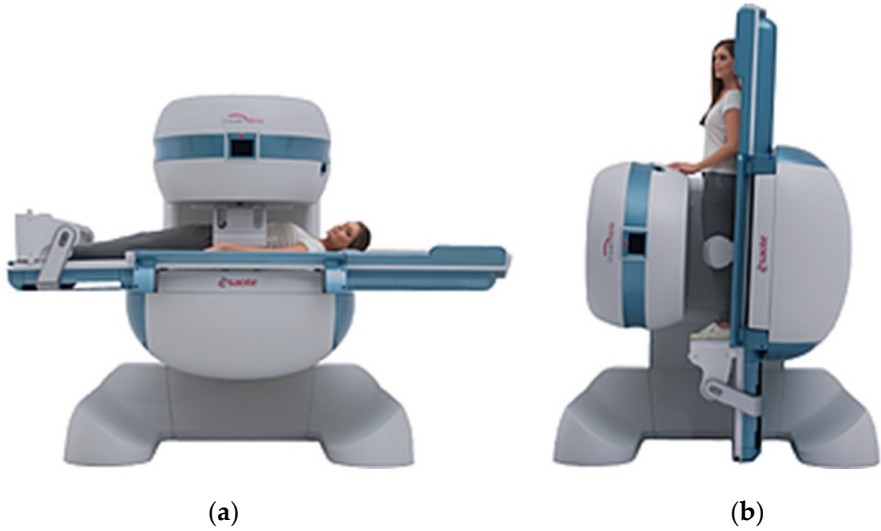

**Figure 1.** The employed MR scanner in traditional (**a**) and weight-bearing (**b**) configuration.

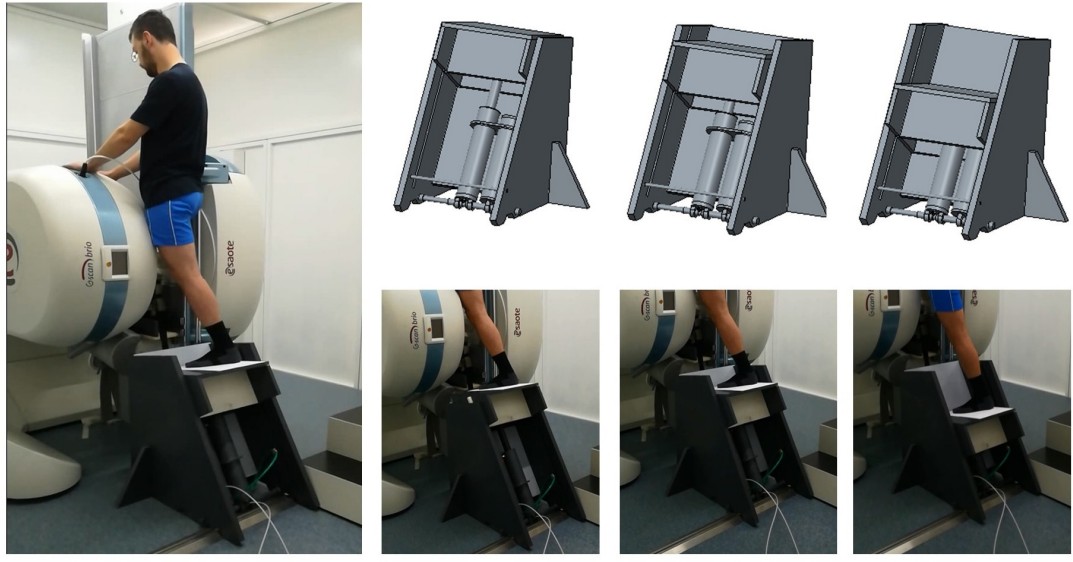

**Figure 2.** The MR-compatible rig used to guide the knee flexion.

The 2D/3D registration algorithm developed for this study is based on voxel intensity, exploiting the low signal associated with the bone tissue within the MR scans to register 3D bone models on the DMRI images through rigid rototranslations. To this aim, a model of the inner bone, namely the internal surface of the cortical and subchondral bone, was also segmented from the non-weight-bearing MRI with an offset of 0.5 mm. Thanks to this offset, when the model is correctly registered on MRI data, for each image, the intersection between the inner bone model and each DMRI plane will take place inside the cortical bone region, which should correspond to the minimum signal intensity on the DMRI images.

Optimal registration is performed within a proprietary C++ code. Bones are moved within the MR reference system, and the intersection between the DMRI planes and stl model is computed for each pose. Since DMRI has a non-zero thickness, all the stl points whose distance to the scanning plane is less than half the slice thickness are considered as belonging to the intersection and projected on the scanning plane. For each intersection point, the corresponding intensity is computed by bilinear interpolation of the DMRI voxel values. For each frame, optimal bone position is obtained by minimizing the mean of this intensity value extended on the overall intersection between DMRI planes and bone stl model.

The registration process just presented is fully automatic; however, it is affected by the initial registration, i.e., the first guess of the first joint pose. To minimize the impact of the operator, initialization of the registration is partially automatized in a separate step. The first DMRI frame of each plane is processed by means of a Channy edge detection algorithm to identify the bone contours, resulting in a cloud of points. The operator is then requested to refine the detected edges, manually eliminating points that do not correspond to the cortical bone. The operator manually registers the bone stl models to this cloud of points. This initial registration is then refined by means of an ICP algorithm and passed, as a starting point, to the 2D/3D intensity-based registration algorithm. A schematic representation of the overall code workflow is given in Figure 3.

### 2.4. Identification of the Optimal Scanning Planes and Registration Accuracy

In order to provide a reference motion to test the registration algorithm performance, five additional static scans (3D HYCE) were also taken in weight-bearing configuration, setting the flexion angle, by means of the rig, approximately at 0°, 15°, 45°, 75°, and 90° (Figure 4). Bones were segmented from all the scans, and the femur, tibia, and fibula from non-weight-bearing MRI were registered to the corresponding bones on each scan through an Iterative Closest Point (ICP) algorithm developed in Matlab (Figure 4). Since the anatomical reference frames were defined on the non-weight-bearing MRI, as noted above, in this way, it was possible to define the rototranslational matrixes describing the relative pose of the anatomical reference systems of the femur and tibia at the five static scans. The femoro-tibial motion was derived by parametrizing the rototranslational matrixes, using the center of the femoral reference system to track the translations and the ZXY cardanic angle sequence to represent the rotations [17].

To reduce the scanning time, the number of DMRI planes acquired for each exam was limited to two. To determine which plane combinations would allow for the optimal motion reconstruction, DMRI was simulated from the five static weight-bearing scans of one volunteer by resampling the original MRI in different planes (Figure 5). The set of tested plane pairs is reported in Table 1. For each combination of planes and each static scan, the registration algorithm was run, and the positional and orientational accuracies were evaluated as the mean absolute error (MAE) between the reconstructed and measured tibio-femoral motion. Once the combination resulting in the smallest rotational and translational MAE was identified, plane orientation was further adjusted to minimize the chance of out-of-plane motion during knee flexion. Accuracy was also tested for these optimal planes to ensure no quality loss in bone registration.

Once the optimal scanning planes were determined, DMRI results were simulated from the static scans for all the three volunteers, as described above, and the registration algorithm was run. With respect to the real dynamic imaging described below, the bone spatial pose is known in this case, thus facilitating an accurate validation of the registration algorithm. The overall rotational and translational algorithm accuracies were then defined as the MAE between reconstructed and measured motion, as well as averaged on the three volunteers.

### 2.5. Dynamic Imaging

For each volunteer, three exams were performed for recording DMRI (2D hybrid contrast enhancement, FOV 200 × 200, pixel spacing 0.68/0.68, slice thickness 5 mm, TR = 20 ms, TE = 10 ms, flip angle 80°, 2.9 s per frame) of the right leg on the two optimal planes. Each exam required two flexions since the G-Scan Brio allows the acquisition on a single plane at a time. To minimize the variations between subsequent acquisitions, DMRI were taken one after the other using the same step velocity. An additional support was introduced to keep the shank fixed in the scanner during the tests while, at the same time, allowing an unconstrained motion at the knee.

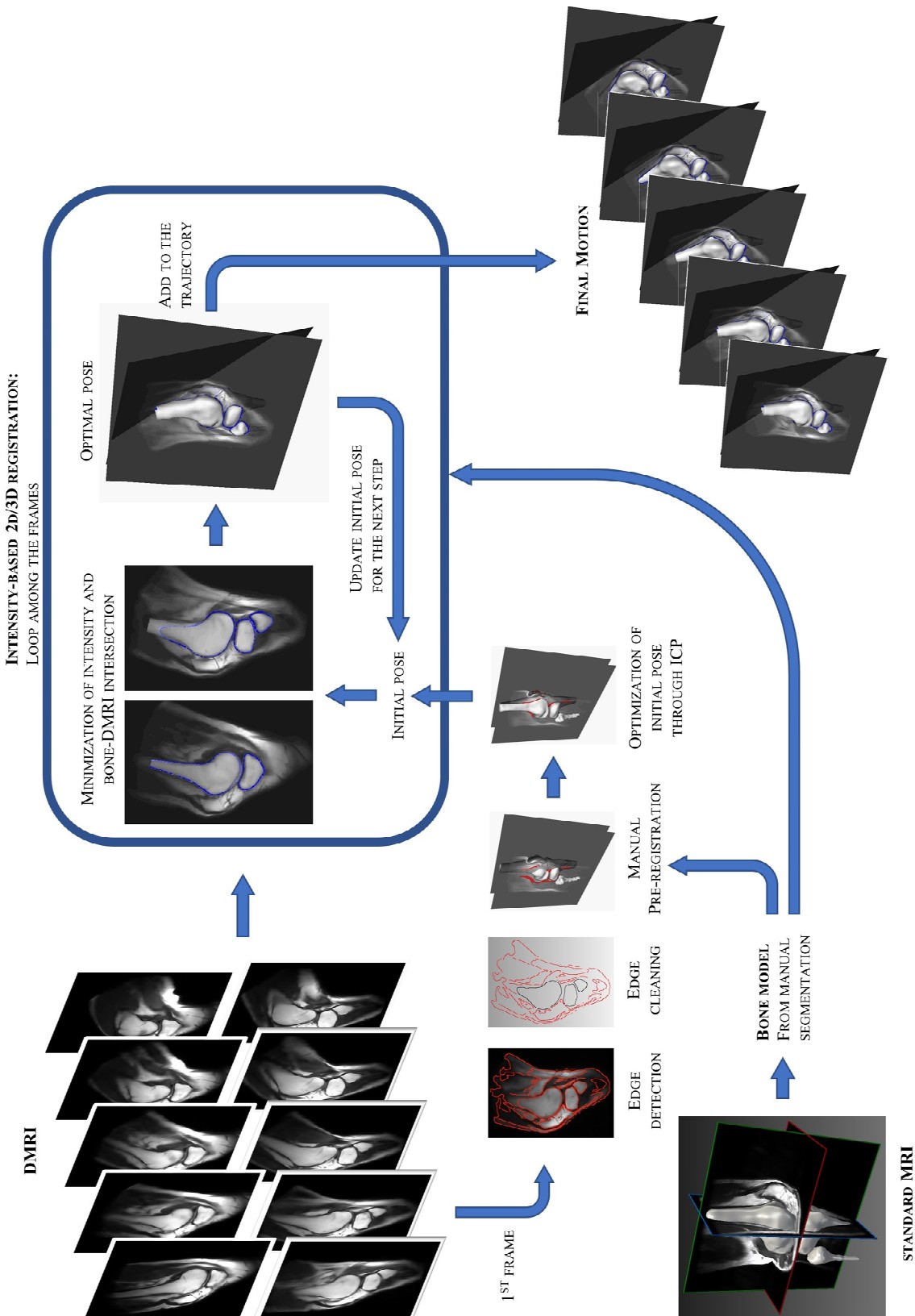

**Figure 3.** Schematic representation of the registration process to the reconstruction of the knee kinematics from DMRI.

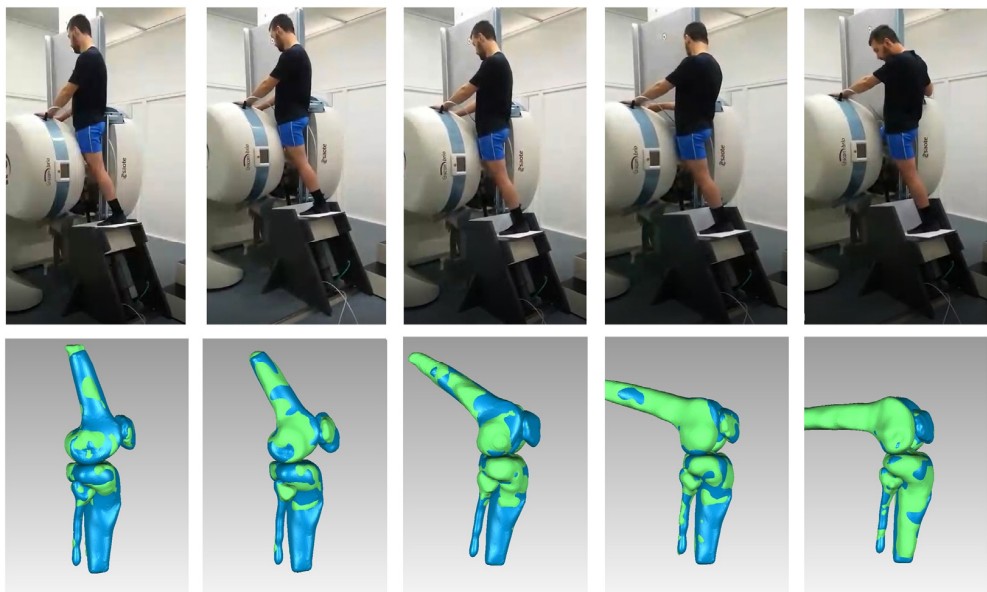

**Figure 4.** Reconstruction of knee kinematics from five static scans at different knee flexion angles: patient position for each scan (**top row**) and corresponding segmentation of knee bones (**bottom row**). The bones from non-weight-bearing scan are represented in green, registered on the corresponding bones, as segmented from each weight-bearing MRI, in blue.

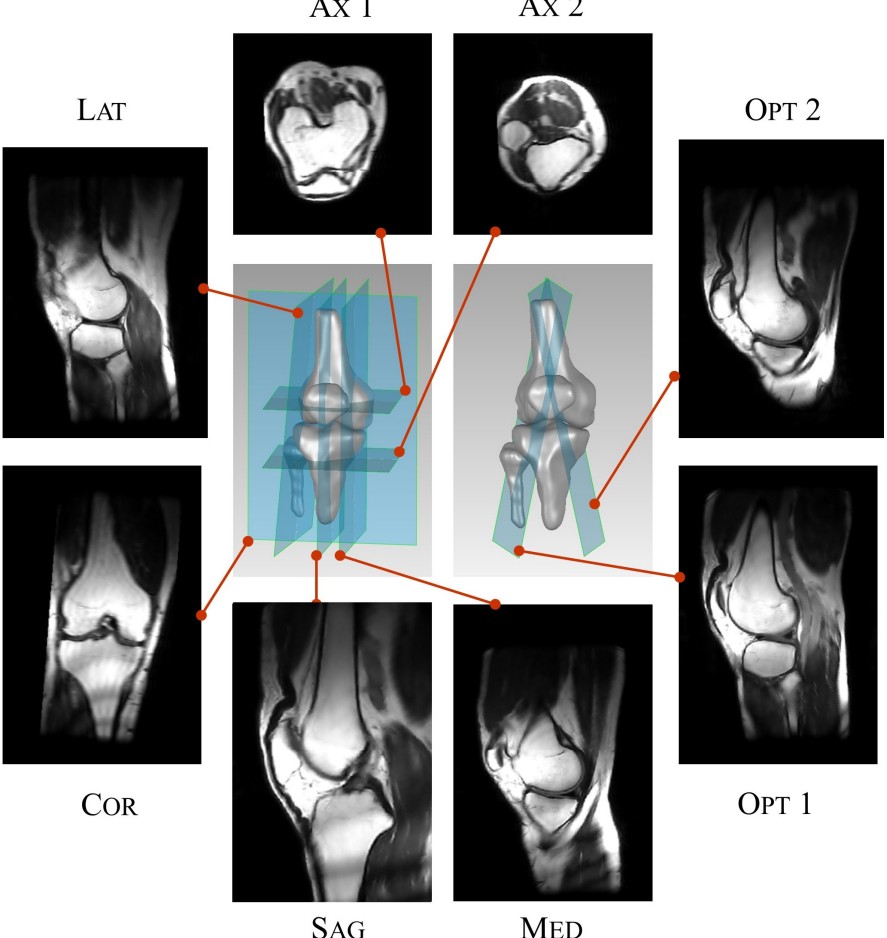

**Figure 5.** Location of the tested DMRI plane for the optimization of the kinematics reconstruction.

The first two exams were identical, while during the third, a wooden block was positioned below the left foot to increase the starting flexion angle to the volunteer's maximum. For each repetition, 71 frames were recorded.

*2.6. Joint Kinematics Reconstruction: Repeatability of the Exam and Sensitivity to the Initial Registration*

The registration algorithm was applied to the three repetitions of DMRI performed by each volunteer. To minimize the effect of the initial registration, simulations were run from full extension to flexion, and then, the frame order was inverted to simulate extension. Only the extension cycle was considered.

To test the repeatability of the experimental procedure, the standard deviation among the three repetitions was computed over the common flexion range for each motion component. Rotational and translational repeatability were defined as the mean standard deviation values over all volunteers.

To test the impact of the initial registration on the final reconstructed motion, the algorithm was run by perturbing the pose of the tibia, fibula, and femur (the three considered as a single rigid complex) first by $\pm 5$ and then by $\pm 10$ (mm and $°$), on each component, for a total of 1456 combinations around the initial registration proposed by the operator. In this case, simulation from full extension to flexion and then back to extension was also run, and only the extension cycle was considered and compared with the motion obtained without perturbation of the initial pose. Trajectories with rotational MAE below $0.5°$ and translational MAE below 0.5 mm were considered not affected by the considered perturbation.

## 3. Results

The rotational and translational MAE for the different pairs of tested planes are reported in Table 1, considering this measure as an indicator of the system accuracy. The optimal planes show the lowest errors.

The registration accuracy estimated on the simulated DMRI for the three subjects was $1.8° \pm 1.4$ and 1.2 mm $\pm 0.8$, for rotations and translations, respectively. In Figure 6, the bone registration on the two DMRI planes is depicted.

Figure 7 shows the comparison among knee motion as reconstructed from a real DMRI exam and as estimated through static scans for the three volunteers, while Figure 8a shows the three repetitions for one volunteer. The overall repeatability for the three volunteers was $3.2°$ and 1.3 mm. Over the 1456 perturbations of the initial registration, 97.6% (1421) resulted in differences below $0.5°$ and 0.5 mm with respect to the unperturbed motion (Figure 8b). In the remaining 2.4% of cases (35), the reconstruction error was significantly detectable, resulting in average rotational and translational differences of $17.3° \pm 7.7$ and 9.7 mm $\pm 4.5$, respectively.

## 4. Discussion

We presented an experimental-computational procedure for the in-vivo quantification of the knee kinematics under physiological loads by means of non-invasive DMRI. The procedure combines a weight-bearing, open MR scanner, a MR compatible rig to guide the knee flexion, and a registration algorithm to reconstruct the motion from DMRI. The procedure is non-invasive and, except for some initialization parameters, completely automatic.

**Table 1.** Rotational and translational MAE for each of the tested combinations of DMRI planes.

| Plane Combination | Rotational MAE [°] | Translational MAE [mm] |
|---|---|---|
| Sag-Cor | 1.8 ± 1.5 | 1.8 ± 1.7 |
| Sag-Ax1 | 2.8 ± 2.4 | 1.8 ± 1.4 |
| Sag-Ax2 | **1.0 ± 0.7** | 1.4 ± 0.9 |
| Sag-Med | 2.4 ± 1.4 | 1.6 ± 1.4 |
| Sag-Lat | 1.8 ± 1.1 | 1.3 ± 1.5 |
| Cor-Ax1 | 3.2 ± 4.6 | 4.2 ± 4.8 |
| Cor-Ax2 | 3.0 ± 3.0 | 2.4 ± 1.9 |
| Cor-Med | 2.8 ± 2.5 | 1.7 ± 1.6 |
| Cor-Lat | 2.5 ± 2.1 | 1.9 ± 1.7 |
| Ax1-Ax2 | 3.1 ± 2.3 | 1.4 ± 0.8 |
| Ax1-Med | 2.2 ± 1.9 | 1.4 ± 1.4 |
| Ax1-Lat | 3.5 ± 1.9 | 5.5 ± 5.8 |
| Ax2-Med | 5.0 ± 3.2 | 2.7 ± 2.7 |
| Ax2-Lat | 2.3 ± 1.5 | 2.8 ± 2.5 |
| Med-Lat | 1.8 ± 1.1 | **1.2 ± 0.6** |
| **Opt1-Opt2** | **1.0 ± 0.5** | **0.7 ± 0.4** |

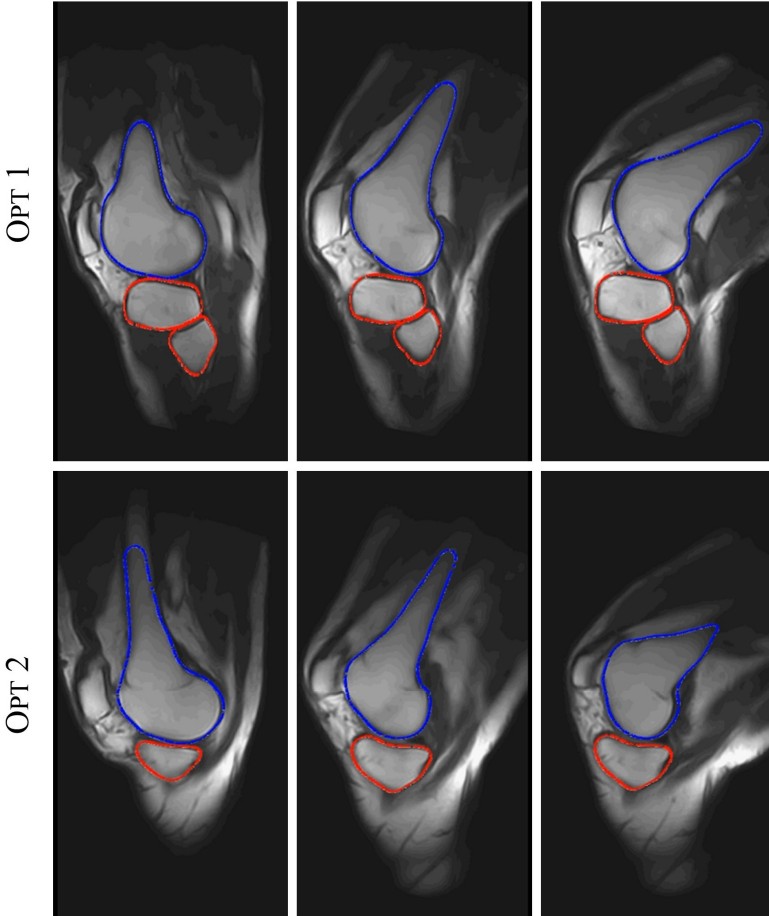

**Figure 6.** Representation of the bone registration, on the two optimal DMRI planes, at different flexion angles. The registered femur intersection with the planes is depicted in blue, while the tibia–fibula complex intersection is depicted in red.

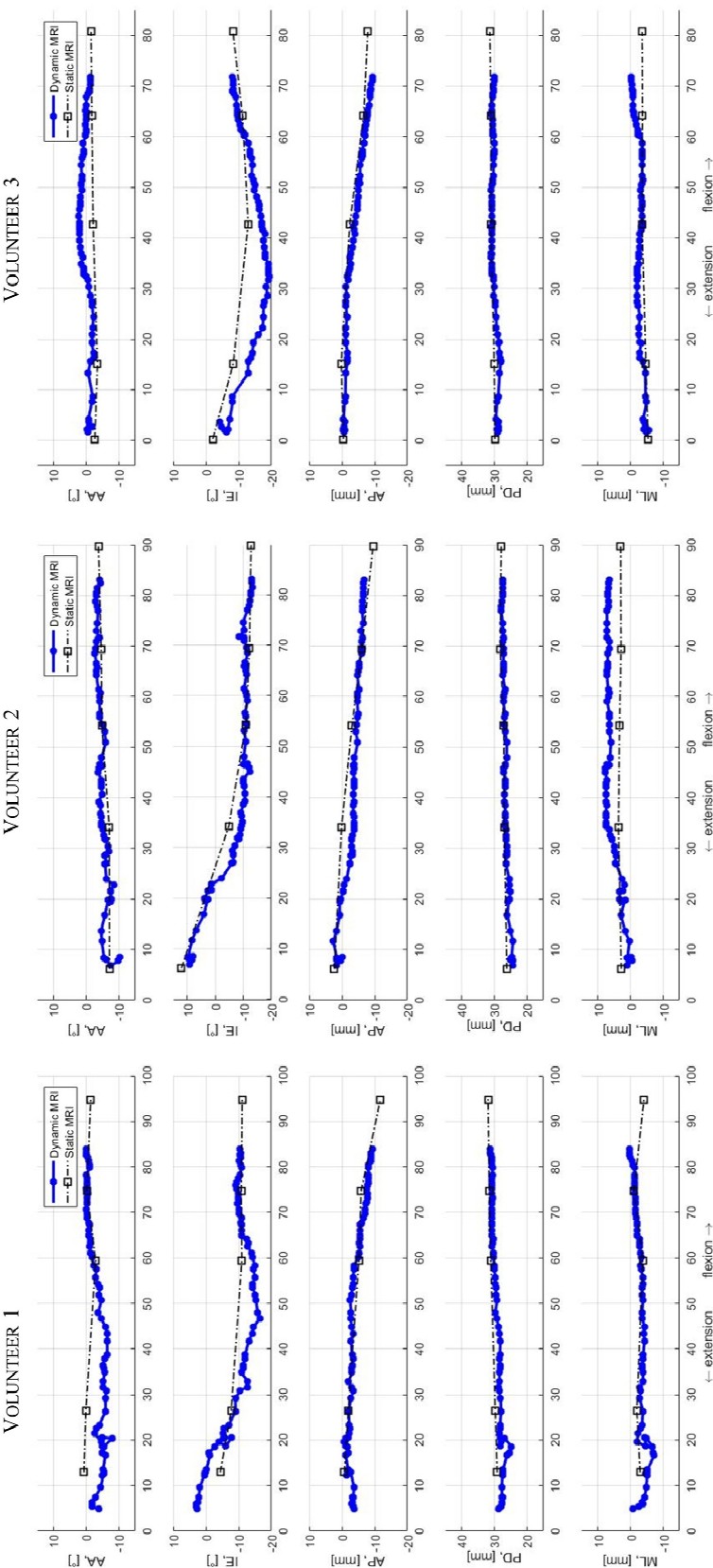

**Figure 7.** Comparison between knee kinematics reconstructed from dynamic (blue) and static (black) MRI. The Abduction/adduction (AA), internal/external rotation (IE), anterior/posterior (AP), proximal/distal (PD), and medial/lateral (ML) translations are plotted vs. the knee flexion angle.

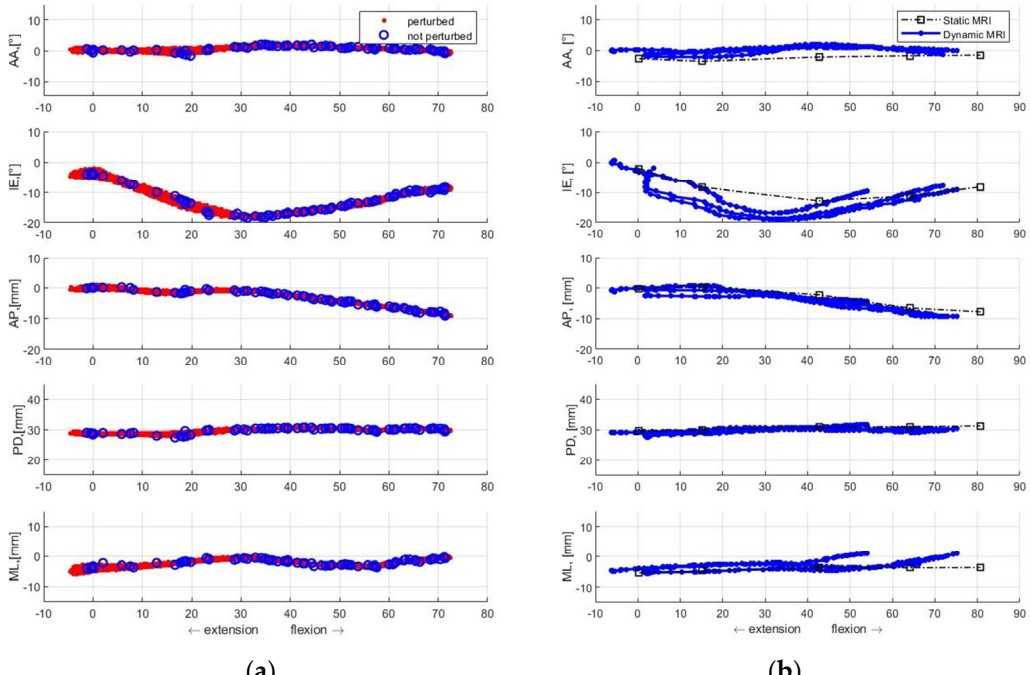

**Figure 8.** (**a**) Comparison between knee kinematics reconstructed from three repetition of dynamic (blue) and static (black) MRI for one volunteer. The Abduction/adduction (AA), internal/external rotation (IE), anterior/posterior (AP), proximal/distal (PD), and medial/lateral (ML) translations are plotted. (**b**) Comparison among the kinematics reconstructed from DMRI without (blue) and with (red) perturbations. In red, the 1421 cases (97.6%) not affected by perturbation of the initial pose are plotted.

The specific scanner employed in this study offers several advantages. Being open, it allows a considerable range of motion. Moreover, the possibility to scan the person in vertical position allows for the analysis of the articulation under the action of the weight and the muscles, thus resulting in a more physiological loading condition.

The 2D/3D registration algorithm shows a rotational and positional accuracy below 2° and 1.5 mm, where the latter is in the order of magnitude of twice the in-plane dimension of the DMRI voxel for this study. The registration accuracy is reasonably limited by the low intensity of the magnetic field of the employed scanner. It has, indeed, been shown that motion tracking through DMRI is proportional both to the strength of the magnetic field and to the velocity of the tracked object [18]. Other studies investigating the knee through DMRI achieved higher accuracy using higher intensities [15]. Nevertheless, the achieved accuracy is still enough to allow for the characterization of the physiological and pathological knee kinematics, while the low intensity of the magnetic field makes the analysis practically non-invasive and compatible, to some extent, with measures on patients with articular prostheses or other small medical devices. The registration algorithm proposed here is general and, thus, extendable to other DMR scanners, reasonably resulting in a higher accuracy.

The registration approach proved to be almost insensitive to the initial pose provided by the operator. In the few cases in which the algorithm did not converge on the reference motion, the results were evidently wrong and not physiological, thus allowing the easy identification of possible errors. Finally, the experimental procedure shows a good repeatability, allowing longitudinal investigations.

The reconstructed knee kinematics agree well with previous measurements [19,20]. In particular, the tibia internal rotation and the femur roll back associated with flexion are easily observable for all the volunteers. The comparison between dynamic and static evaluation of knee kinematics shows differences smaller than what was reported in the

literature [7]. It is worth noting that static scans were collected in weight-bearing conditions and that dynamic scans were recorded at slow speed, possibly reducing the differences, although part of them could be ascribed to the different measuring conditions. In general, however, it is possible to observe a reduction in the maximum flexion reached during dynamic measurements, which is possibly associated with adduction of the pelvis during static scans. It is worth noting that the maximum flexion value was dictated by the possible stroke of the hydraulic piston in the rig, which was kept above a safety value for this investigation. Future tests will extend the maximum achievable flexion.

The clinical application of DMRI could provide complementary information to what is obtainable with traditional MRI. The latter provides very accurate yet static images of the joint structures; on the other side, DMRI allows for the observation of the interaction of the elements that participate in an articulation during its function, providing a new level of knowledge. For example, laxity tests could be performed in DMRI, making it possible to directly observe the load response of injured ligaments. In general, a quantification of the relative bone motion makes it possible to find the measure of quantities not directly observable in-vivo. For example, the amplitude and location of articular contact areas during knee flexion can be reconstructed from the tibio-femoral kinematics, providing data that may help to better understand the etiology and development of pathologies such as osteoarthritis.

Aside from the direct clinical applications, this procedure for the in-vivo quantification of knee joint kinematics has very interesting biomechanical applications. Indeed, the possibility to measure the individual joint kinematics non-invasively and in-vivo will help in the definition and validation of patient-specific joint and musculoskeletal models [21–25].

The work has limitations. Acquisitions on the two DMRI planes used for 2D/3D registration were performed in a series. It is thus possible that the two acquired motions differ, introducing some errors in the reconstructed motion. The same kind of approximation is, however, done with traditional cine-MRI, where a cyclic motion is reconstructed from successive scans taken at different times in subsequent cycles. Only three knees were analyzed in this investigation. A wider study will establish the performance of the proposed procedure.

Future work will test the presented algorithm on data from a 1.5 T MR scanner. Other anatomical compartments will be also investigated.

**Author Contributions:** Conceptualization, M.C., N.S. and V.P.-C.; methodology, M.C. and F.D.C., N.S. and G.M.; data collection, M.C., F.D.C., N.S. and G.M.; software, M.C. and M.B.; validation, M.C. and M.B.; formal analysis, M.C., M.B. and N.S.; writing—original draft preparation, M.C., M.B. and N.S.; writing—review and editing, M.C., F.D.C., M.B., N.S., V.P.-C. and G.M. All authors have read and agreed to the published version of the manuscript.

**Funding:** This research received no external funding.

**Institutional Review Board Statement:** Not applicable.

**Informed Consent Statement:** Informed consent was obtained from all subjects involved in the study.

**Data Availability Statement:** Not applicable.

**Conflicts of Interest:** The authors declare no conflict of interest.

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
