# Peer review of "In-Vivo Quantification of Knee Deep-Flexion in Physiological Loading Condition trough Dynamic MRI"

_applsci, doi:10.3390/app13010629_

Round 1

Reviewer 1 Report

Interesting article on novel procedure that reconstructs 3D knee kinematics using MRIs and registration algorithm created in the study.  The methods appear to be sound. Results of verification of the newly created registration algorithm are provided, which greatly strengthens the study. However, important parts of description of the methods lack clarity and the fact that only one volunteer has been analysed is an important limitation in terms of practical evaluation of the proposed approach.

Detailed comments and questions are below.

1)     Page 4: “To this aim, a model of the inner bone, namely the internal surface of the cortical and  subchondral bone, was also segmented from the non-weightbearing MRI…”

a)    Does it mean that both inner bone and subchondral bone were segmented?

b)    Does this statement imply that the proposed registration algorithm relies on/requires bone segmentation?

c)    Registration typically involves a transform that “warps” source image onto the target image. Does the proposed registration algorithm employ rigid body/rigid transform (i.e. rotations and translations) or other type of transform.

2)     Page 7: “The registration accuracy estimated on the simulated DMRI for the three subjects was 1.8°± 1.4 and 1.2 mm ± 0.8”

a)     Should the statement that the accuracy was 1.8°± 1.4 and 1.2 mm ± 0.8 be interpreted as that the maximum registration error was 3.2° for angular displacement and 2 mm for position/translation?

b)     How does the error in position/translation compare to voxel size of the MRIs used in the study?

c)     Is the reported accuracy sufficient from the perspective of the targeted application: understanding of the joint natural behavior and the comprehension of articular disorders.

d)     What is the cost of the MRI scanner used in the study? How feasible would be wide application in clinics in analysis of articular disorders.

3)     Quality of MRIs tends to vary between the acquisitions, patients, and the scanners used. Evaluation/demonstration of the proposed approach has been conducted against the images of only one volunteers acquired using the same scanner. This demonstrates potential of the approach, but provides only limited evidence of the approach performance. I believe that this limitation should be discussed in the manuscript.

4)     Analysis (“comprehension”) of articular disorder is one of the key motivations for the study. However, no example of application of the proposed approach in such analysis is presented in the manuscript. There is even no specific discussion of how the proposed can be applied in the articular disorder analysis. I believe that such discussion should be included in the manuscript.

5)     Page 4: “performances” — do you mean “performance”?

6)     Page 4: “resampling original MRI on different planes” — “on different planes” or “in different planes”?

Reviewer 2 Report

What is the main question addressed by the research?

This is not a question driven nor hypothesis driven paper. The authors do not pose any research question and they do not formulate any research hypothesis. They do, however, define an objective which is to provide a novel alternative to the current procedure of taking dynamic MRI to analyze the internal structure of the knee joint during movement. Instead of the traditionally employed horizontal orientation of the scanning bed, they propose a vertical, weight-bearing configuration. This allows a simpler and more natural loading of the knee joint under investigation thus making it possible to better mimic the physiological load conditions experienced during stair climbing or during walking.

If a guiding question were to be filtered out of the text of the manuscript, that would probably be whether the newly proposed procedure can provide sufficient accuracy for a clinically useful description of the knee joint behavior under natural, dynamic loads.

Do you consider the topic original or relevant in the field? Does it address a specific gap in the field?

Although it is not the aim of the manuscript to provide new data on knee joint structures under dynamic loads, it provides a fairly detailed description of a novel experimental procedure which may potentially result in new findings. It solves the problem of imaging the knee joint under loads that correspond exactly to those experienced in real life situations, like in locomotion.

What does it add to the subject area compared with other published material?

It provides a new, better way of performing dynamic MRI of joints by bringing the possibility of loading them more naturally, using the body weight.

What specific improvements should the authors consider regarding the methodology? What further controls should be considered?

The authors should consider increasing the number of subjects (knees) participating in the tests.

Are the conclusions consistent with the evidence and arguments presented and do they address the main question posed?

Conclusions are not explicitly formulated. However, one can infer them from the Results and Discussion chapters. The authors claim the accuracy achieved when employing the proposed procedure is sufficient to envisage direct clinical applications and development of patient-specific biomechanical models. In this way they address the de facto main question, even though they do not express it explicitly.

Are the references appropriate?

Yes, the references are up to date and appropriate.

Please include any additional comments on the tables and figures.

Figures 7 and 8 are considerably more difficult to read than the remaining ones.

I would like to commend the authors for preparing this important and high quality manuscript. I have only a few minor suggestions on how to improve the text. They are listed below.

Line 35, what measured --> what is measured

line 45, side --> (?) hand

line 60, possibility --> (?) visibility

line 71, what experienced --> what is experienced

line 102, corresponding --> (?) correspond

line 120, points cloud --> cloud of points

line 129, trough --> through

line 218, except some --> (?) except for some

line 253, to measures --> with measures

line 254/255, here proposed --> (?) proposed here

line 265, what reported --> what is reported

Reviewer 3 Report

Authors of paper „In-vivo quantification of knee deep-flexion in physiological loading condition trough dynamic MRI” address the topic of dynamic imaging of knee joint with use of MRI.

Very strong part of paper is its abstract. It is very well written and provide all information about work and motivation of authors to perform those investigations .

Authors wrote very good introduction, justifying why they do their research. Results are written very laconically and could be a little bit expanded basing on figures provided by authors.  Discussion is written in good way nevertheless some information about significance of authors results in the context of particular diseases diagnosis is missing.

Major:

1.       Authors must increase quality of figures. They are pixelized. It is imposible to see what is on Figure 8. Please use 300 or even 600 dpi.

2.       At page 4, line 129 authors refer to ICP algorithm in Matlab. One of drawback of its drawback is fact that Matlab is not an open source software. Could authors provide open-source alternatives for their data analysis pipeline?

3.       Authors clearly indicate limitation of their study. Nevertheless this study could benefit from including volounteers with diagnosed musculoskeletal diseases to assess if method proposed by authors is better in diagnosis that static MRI. How authors could refer to this issue?

4.       Please provide at least how particular parameters which you can measure with your approach in contrast to static measurement could be used for particular musculoskeletal diseases diagnosis.

Minor:

1.       Authors refer that musculoskeletal disorders are the second most common cause of disability world-wide. They could refer what is first.

2.       At page 1, lines 36-38 authors claim “several studies” while referring only to one manuscript. Please consider referring to more direct studies or rewrite this statement.
